

# Continuous observation of Stable Isotopes of Water
# Vapor in Atmosphere Using High-Resolution FTIR
Chang-gong Shan [1, 2], Wei Wang [2*], Cheng Liu [2,3,4*], You-wen Sun[2], Yuan Tian[2], Isamu
Morino[5]
[1]School of Environment science and Optoelectronic Technology, University of
Science and Technology of China, Hefei, 230000, China
[2] Key Laboratory of Environmental Optics and Technology, Anhui Institute of Optics
and Fine Mechanics, Chinese Academy of Sciences, Hefei, 230031, China
[3] University of Science and Technology of China, Hefei, 230000, China
[4]Center for Excellence in Urban Atmospheric Environment, Institute of Urban
Environment, Chinese Academy of Sciences, Xiamen, 361021, China
[5]Satellite Observation Center, National Institute for Environmental Studies, Tsukuba,
305-8506, Japan
Correspondence to: Cheng Liu (chliu81@ustc.edu.cn),
Wei Wang (wwang@aiofm.ac.cn)
**Abstract**
Observations of stable isotopes of water vapor provide important information for water
cycle. The volume mixing ratios (VMR) of $H_2O$ ($X_{H2O}$) and HDO ($X_{HDO}$) have been
retrieved based on a high-resolution ground-based Fourier transform infrared
spectroscopy (FTIR) at Hefei site, and the isotopic composition δD was calculated.
Time series of $X_{H2O}$ were compared with the Greenhouse gases Observing Satellite
(GOSAT) data, showing a good agreement. The daily averaged δD ranges from -17.02‰
to -282.3‰ between September 2015 and September 2016. Also, the relationships of
meteorological parameters with stable isotopologue were analyzed. δD values showed
an obvious positive correlation with temperature and $\ln(X_{H2O})$ and a weak correlation
with relative humidity. Further, 51.35% of airmass at Hefei site comes from the
southeast of China, and the main potential sources of δD are in the east of China over
the observation period based on the back trajectories model. Furthermore, the δD values
of evapotranspiration were calculated based on Keeling plot. Observations of the stable
isotopes of water vapor by high-resolution ground-based FTIR provide information on
study of the variation of the atmospheric water vapor at Hefei site.
**1. Introduction**
Water cycle plays an important role in climate change. Water vapor plays a key role in





cloud formation progress, however, its associated feedback mechanism is poorly known
(Soden et al., 2005; Boucher et al., 2013). Observations of stable isotopes of water
vapor in the atmosphere provide important information for hydrological cycle, because
the stable isotopes change with the phase change of water vapor. The variation of stable
isotopes of water vapor in the atmosphere reflects the change of water cycle, and the
measurements of stable isotopes reveal the relationship between atmospheric dynamics,
evaporation, and condensation process (Yoshimura et al., 2008; Risi et al., 2010).
The stable isotopologues of water vapor mainly include $H_2^{16}O$, HDO and $H_2^{18}O$. The
HDO/$H_2O$ ratio is usually expressed as a ratio of HDO to $H_2O$ abundance. The "delta
notation" is usually used to represent the isotopic composition, and normally defined
as:

$$\delta D = (\frac{R_m}{R_s} - 1) \times 1000‰ \qquad (1)$$

Where $R_s$ (equals to $3.1152 \times 10^{-4}$) is the standard HDO abundance of Vienna standard
mean ocean water (VSMOW), and $R_m$ is the measured ratio of HDO/$H_2O$ (Craig et
al., 1961).
Water vapor mainly exists in the troposphere, more than 60 % of water vapor are below
850 hPa and 90 % below 500 hPa (Ross et al., 1996). Gribanov (2014) proved that the
column averaged HDO/$H_2O$ ratio is highly correlated with near surface $\delta D$. Recent
studies used column averaged HDO/$H_2O$ ratio combined with in-situ $\delta D$ measurements
to study the seasonal and inter-seasonal variations of water cycle (Gribanov et al., 2014).
The variation of atmospheric temperature and humidity near the surface also cause the
atmospheric water recycling (Boucher et al., 2004; Destouni et al., 2010; Tuinenburg et
al., 2012). Therefore, many studies reported that meteorological parameters at ground
level are correlated with the stable isotopologue of water vapor. For example, $\delta D$ have
a positive correlation with temperature and relative humidity of the atmosphere in
summer in Mediterranean coastal area (Delattre et al., 2015). Bastrikov (2014) also
analyzed the relationship between $\delta D$ and temperature and humidity in different
seasons in West Siberia. However, these reports are based on in-situ measurements, and
there are few studies about the relationship between the column averaged HDO/$H_2O$

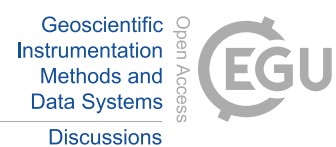

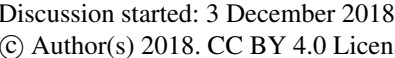

ratio δD and the meteorological parameters.
Ground-based FTIR technique is widely used to obtain long-term time series of
atmospheric composition and validate satellite data (Schneising et al., 2012;
Scheepmaker et al., 2015). High-resolution FTIR observations have achieved accurate
detection of greenhouse and trace gases (Washenfelder et al., 2006). The Total Carbon
Column Observing Network (TCCON) and the Network for the Detection of
Atmospheric Composition Change (NDACC) use high-resolution FTIR instrument to
accurately and precisely derive the main stable isotopologue of water vapor, HDO
(Hannigan et al., 2009; Wunch et al., 2011). The total column of HDO and $H_2O$ are
retrieved in the near infrared region, and the column averaged $HDO/H_2O$ ratio are
calculated. Also, the Column averaged HDO derived from the high-resolution FTIR
instrument have been used for comparison with model simulations and satellite data
(Boesch et al., 2013; Frankenberg et al., 2013; Rokotyan et al., 2014; Dupuy et al.,

77  2016).

Water isotopologues composition has been analyzed in Hefei with an obvious seasonal
variation, only at the month scale, using in situ measurements (Wang et al., 2012).
However, so far no research has been dedicated to the water vapor and its isotopologues
variation in a large spatial-temporal scale at Hefei. To better understand
evapotranspiration, process and the relationship between meteorological parameters
and water vapor isotopologues, the column stable isotopologues of water vapor
observed by ground-based FTIR technique are presented in the paper.
The instrumentation and retrieval strategy for column averaged $H_2O$ and HDO at Hefei
site are described in Section 2. The retrieval results are discussed in Section 3, also, the
relationships between the isotopic composition δD and temperature, relative humidity
are analyzed. Moreover, the evapotranspiration signature $δ_{ET}$ and the sources of water
vapor based on the back trajectories calculation of air masses are clarified in this
Section. The conclusions are given in Section 4.
**2. Instrumentation and retrieval strategy**
The ground-based high-resolution FTIR spectrometer (Bruker IFS 125 HR) and solar



tracker (A547) installed on the roof of laboratory, are combined to collect the solar
absorption spectra at Hefei site. Hefei (31.9 °N, 117.17 °E, about 30 m above the sea
level) is a continental site, away from the southeast urban area about 10 km (Figure 1).
The $CaF_2$ beamsplitter and InGaAs detector are used to collect the near-infrared (NIR)
spectra. The NIR spectral range covers 4000-11000cm$^{-1}$, and the spectral resolution is
0.02 cm$^{-1}$, corresponding to a 45 cm maximum optical path. In order to ensure the
stability of the measurement, the instrument is vacuated under 10 hPa. A weather station
is installed near the solar tracker on the roof of the lab building to record meteorological
data. Wang (2017) described the instrumentation and the measurement routine at Hefei
site.
The solar spectra collected from September 2015 to September 2016 are analyzed. We
use the GGG2014 software package to retrieve the water vapor and its isotopes (Wunch
et al., 2015). GGG2014 is a nonlinear least square spectral fitting algorithum (GFIT),
which scales an a priori profile derived from the National Centers for Environmental
Prediction and the National Center for Atmospheric Research (NCEP/NCAR)
reanalysis data (Toon et al., 2014) to minimize residulas between measured and
simukated spectra. GGG2014 produces the total column of trace gases, then the
column-averaged dry-air mole fractions (DMF) of trace gasees are computed as:
$$X_{gas} = \frac{column_{gas}}{column_{air}^{dry}}$$

$$= 0.2095 \times \frac{column_{gas}}{column_{O_2}} \tag{2}$$

The column of dry air, units of molecules/cm$^2$, is computed from the oxygen ($O_2$)
column (Wunch et al, 2011) dividing by 0.2095. Figure 2 depicts the spectral fitting of
the $H_2O$ and HDO in the spectral window of 4565-6470 and 4054-6400 cm$^{-1}$,
respectively. The rms spectral fitting residuals are 0.16% and 0.25% for $H_2O$ and HDO
respectively. Table 1 lists the spectral windows for column retrievals of $H_2O$ and HDO,
which are the standard GFIT windows. Figure 3 shows the column averaging kernals
of $H_2O$ and HDO. The difference of the column averaging kernals below 500 hPa
between them is very small, with the value of 4.34%.



## 3. Results

### 3.1. Time series of δD, water vapor and meteorological parameters

The DMFs of $H_2O$ and HDO are calculated using total columns of $H_2O$ and HDO based on equation (2). The δD time series at Hefei station is plotted in Figure 4 from September 2015 to September 2016. The precision of δD (1-σ precision divided by the measured value) is about 3.63%. The daily averaged δD varies from -17.02‰ to -282.3‰. δD shows an obvious seasonal variation over the observed period, with the lowest δD values occurring in mid-January and the peak in early August.

The time series of $X_{H2O}$ and meteorological parameters from September 2015 to September 2016 at Hefei station are plotted in Figure 5. The mean relative retrieval error (1-σ precision divided by the measured value) of $X_{H2O}$ is about 1.11%. The variations of $X_{H2O}$ are similar to those of δD, with an obvious seasonal pattern. The variation of $X_{H2O}$ is large during the period. The daily averaged $X_{H2O}$ was in the peak of 8821.97 ppm in early August in summer and reduced to the minimum of 225 ppm in mid-January in winter. The variation of surface temperature is close to $X_{H2O}$ variation, while the relative humidity of atmosphere shows a weak seasonal variation. The peak and valley values of water vapor and δD seem to accompany with those of temperature, and the different amplitude of daily variation of δD in different seasons depends on temperature, therefore, the relationships of water vapor and δD with temperature are discussed in sec.4.2.

## 4. Discussion

### 4.1 Comparison with nearby TCCON observations and satellite data

The time series of $X_{H2O}$ are compared with the GOSAT data (v02.72) from September 2015 to September 2016. For co-locating the GOSAT data with the ground-based FTS data, the GOSAT observations of $\pm 5°$ latitude and longitude centered in the Hefei site within $\pm 2$ hour overpass were selected (Kuze et al., 2009; Yoshida et al., 2013; Scheepmaker et al., 2015). In order to eliminate the influence of different a priori profiles and averaging kernels on $X_{H2O}$, we use a priori profile of the ground-based FTS to correct the column-averaged mole fractions of gases from GOSAT (Reuter et al.,

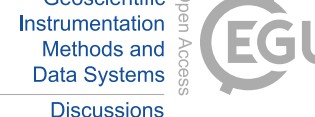

2011; Zhou et al., 2016). The comparison results of $X_{H2O}$ are depicted in Figure 6. The
mean bias, which is defined as the mean difference of $X_{H2O}$ between FTIR and satellite
date, is about 11.98ppm. The $X_{H2O}$ observed by FTIR showed a similar variation trend
with the corrected satellite data, and the variation range agrees with that of GOSAT data.
Since water vapor mainly concentrate in the lower troposphere, and the ground-based
observations have high sensitivity near surface, but the satellite data are insensitive in
the lower troposphere, so the FTIR data are slightly higher than the satellite data. Also,
we calculated the correlation between FTIR and GOSAT data, and there is a high
correlation between FTIR and GOSAT data (R = 0.98). The correlation coefficients
between FTIR and GOSAT data are 0.95 and 0.93 for Japanese Tsukuba and Saga site,
respectively (Dupuy et al.; 2016). The slope of the scatter plot of our FTIR and GOSAT
data is 0.98. It is concluded that FTIR data at Hefei site agree well with the satellite
observations.
Furthermore, to verify the accuracy of our calculated data, we compare the isotopic
ratios δD from Tsukuba TCCON station (Morino et al., 2014) with our δD values.
Tsukuba TCCON station (36.05°N, 140.12°E, 31m above the sea level) is a Japanese
TCCON station close to our site and at a similar latitude (Figure 1). Figure 7 is the plot
of δD in Hefei compared to those of Tsukuba from September 2015 to February 2016.
It is found that the δD in Hefei showed a similar trend as that in Tsukuba, both with the
maximum value in summer and the minimum in winter. During the observation period,
the δD of the two sites began to fall from October 2015 and to the valley value in
January 2016. Hefei and Tsukuba sites have a similar atmosphere circulation pattern
due to the similar latitude, which may result in the similar variation in the stable
isotopes of water vapor in the atmosphere, as shown in Figure 7. However, the daily
averaged δD of Hefei ranges from -36.46‰ to -282.3‰ during this period, while δD in
Tsukuba is from -35.74‰ to -198.37‰, falling in the range of our δD. Scheepmaker
(2015) plots the time series of δD in six TCCON stations, and the δD observed from
these stations in the Northern hemisphere are in the range from about -50‰ to -300‰,
which are comparable to those of our results.



**4.2. Relationship of stable isotopes of water vapor with meteorological parameters**

Atmospheric circulation strongly affects the variations of stable isotopic compositions of water vapor in the atmosphere (Deshpande et al., 2010; Guan et al., 2013). The spatiotemporal distribution of water vapor in the atmosphere is strongly correlated with the weather, and the stable isotopic ratios of water vapor change with the meteorological parameters (Noone et al., 2012, Vogelmann et al., 2015). The surface meteorological data are important for quantifying the distributions of the stable isotopes of water vapor. The statistical data of monthly averaged $\delta D$ and surface temperature are summarized in Table 1. The monthly averaged surface temperature decreased from 30.18 to 4.74 ℃ between Sep.2015 and Jan.2016, and the variation of $\delta D$ also dropped from -126.89‰ to -257.86‰ at the same time. Especially, the daily averaged $\delta D$ reached the minimum of -282.3‰ in 25 January 2016, which is the coldest day during the period. Also, $\delta D$ shows a large variation in winter, with the monthly variation amplitude of 186.38‰ and 213.66‰ in December 2015 and February 2016, respectively. However, the monthly variation amplitude of $\delta D$ in summer is about one third of the corresponding values in winter. Furthermore, the monthly variation amplitude of temperature is 14.1 and 19.2℃ in December 2015 and February 2016, respectively, while the corresponding value is 6.3 and 8℃ in July and August, respectively. It is noted that the correlation coefficient between monthly variation amplitude of $\delta D$ and temperature is 0.95. So it is concluded that the surface temperature strongly influences the variation of $\delta D$ in Hefei site.

For all the data collected, the linear relationship of individual $\delta D$ and the surface temperature is expressed as $\delta D = 5.30‰T - 242.64‰$. The correlation coefficient is 0.83 between $\delta D$ and temperature at Hefei site, as shown in Figure 8(a). Bastrikov (2014) and Bonne (2014) found that there was a positive correlation between the stable isotopes of water vapor and temperature in western Siberia and southern Greenland. In Bastrikov (2014), the slope of $\delta D$ and temperature in western Siberia is 3.1‰℃$^{-1}$. The evaporation of water vapor weakens with the decrease of temperature, and heavier isotopologue, HDO, condenses more actively and evaporate less actively than the main isotopologue $H_2O$ due to their different saturation vapor pressure, so the depletion in heavy isotopes with decreasing temperature happens.

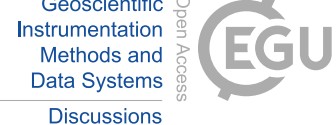

δD of atmosphere in Hefei show a weak correlation with relative humidity, as plotted
in Figure 8(b). The correlation coefficient of linear regression between δD and relative
humidity is 0.45, and the slope of linear regression is 2.11‰%$^{-1}$. Wen (2010) reported
that the stable isotopes of water vapor in Beijing is positively correlated with the
relative humidity (R = 0.42), while the diurnal and seasonal variation of δD have a
strong relationship with the relative humidity in northwest Greenland (Steen-Larsen et
al., 2013).
A simple distillation model, Rayleigh distillation model, helps to understand the
relationship between δD and $H_2O$ (Schneider et al., 2010). The variation of water vapor
and δD are connected via the equation
$$\delta D \times 1000 = (1 + \delta D_0) \times \left(\frac{XH_2O}{XH_2O_0}\right)^{\alpha-1} - 1 \qquad (3)$$
In which $\delta D_0$ and $XH_2O_0$ are the deuterium and water vapor of the airmass from the
ocean, while α represents the fractionation coefficient between the oceanic source and
the sampling site.
There is a linear relationship between ln(δD/1000+1) and ln($X_{H2O}$) , according to the
equation (3). The slope of ln(δD/1000+1) and ln($X_{H2O}$) represents a measure of the
transport pathway of water vapor. Analysis of the slope allows investigating the
importance of different hydrological processes (Worden et al., 2007; Schneider et al.,
2010). As shown in Figure 8(c), there is a strong correlation (R=0.88) between
ln(δD/1000+1) and ln($X_{H2O}$), and the slope of linear regression is 0.081. The results
prove that the stable isotopes of water vapor are highly correlated with the fraction of
water remaining in the cloud. In western Siberia, the correlation coefficient of linear
regression between ln(δD/1000+1)/ln($X_{H2O}$) is 0.71, and the slope of linear regression
is 0.07 (Gribanov et al, 2014).
**4.3. Variation sources of regional δD in Hefei**
The NOAA Hybrid Single-Particle Lagrangian Integrated Trajectory (HYSPLIT)
model is a complete system using NCEP/NCAR reanalysis data to understand transport
paths and sources of air masses (Draxler and Rolph, 2003; Stein et al., 2015). The
HYSPLIT model is used to analyze the Potential Sources Contribution Function (PSCF)



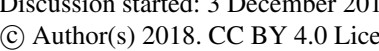

of air parcels (Li et al., 2012). The back trajectories of 72 hours are calculated for each
day, and the height of the backward trajectories is set as 500 magl. The geographic
region precision is selected as 0.5°× 0.5° grid cells in the calculation. The PSCF
calculated by the backward trajectories is weighted according to the method of Polissar
et al. (1999) to identify the source strength (WPSCF).
Figure 9 shows the cluster analysis results and the WPSCF distribution of δD during
the period from September 2015 to August 2016. The sources of air masses of Hefei
area mainly originated from three regions: the Southeast China (SEC), North of China
(NC) and Northwest of China (NWC). 51.35% of airmass were from SEC during the
observation period. Also, The WPSCF analysis indicates that the main potential sources
of δD are near the Hefei site. The potential source of δD are divided into three regions:
the east area with moist and warm airmass, the north area with dry and cold airmass,
and the southwest area with moist and warm airmass. Especially the main airmass from
the east area bring the moist and warm airmass into Hefei, which result in the
enrichment of heavy isotopes.
**4.4 δ-value of evapotranspiration**
Keeling plot is usually applied to estimate the δ-value of evapotranspiration (Keeling
et al., 1958; Wei et al., 2015). The Keeling equation assumes that the actual atmospheric
water vapor is the mixing of the atmospheric background and an additional component
from local evapotranspiration, and each component has distinct isotopic signature. The
water vapor and its isotopes in the atmosphere can be written as (Yepez et al., 2003;
Williams et al., 2004; Sun et al., 2005)

$$\delta_m = (\delta_b - \delta_{ET})W_b \left(\frac{1}{W_m}\right) + \delta_{ET} \qquad (4)$$

Where $W_m$ and $\delta_m$ are DMF and δ-value of the water vapor, respectively. $W_b$ and
$\delta_b$ are DMF and δ-value of the background, respectively. $\delta_{ET}$ is the δ-value of
evapotranspiration. Therefore, the evapotranspiration signature ($\delta_{ET}$) is also expressed
as the y-axis intercept of equation (4).
Keeling plot is used to calculate the δ-value of the evapotranspiration of water vapor.
The days with 4-hour continuous observations are considered to ensure that the data are



representative. The $\delta D$ and $1/X_{H2O}$ have a high-negative correlation in daily timescale,
as shown in Figure 10. The correlation coefficients are -0.97 and -0.85, and the y-axis
intercepts of the linear regression line represent the $\delta D$ from evapotranspiration source
of water vapor, which are -35.39 ‰ and -53.18 ‰ for October 27, 2015 and December
17, 2015, respectively. The time series of $\delta D$ for evapotranspiration obtained from
keeling plot analysis during the measurement period are shown in Figure 11. Over the
period, $\delta D$ value of evapotranspiration varied from (15.3 ± 2.9) ‰ to (-114 ± 8.9) ‰,
and the averaged $\delta D$ value of evapotranspiration is -44.43 ‰. It is seen that the variation
range of $\delta D$ value for evapotranspiration was large, reflecting the fact that the source
isotopic signal did not keep constant over the measurement period. In the study of Wang
(2012), the deuterium isotopic signature from evapotranspiration is between -113.93 ±
10.25 ‰ and -245.63 ± 17.61 ‰ in July in Hefei. Griffith (2006) found that the
deuterium isotopic ratio from evapotranspiration is between -90 ‰ and -100 ‰ in a
pasture.

**5. Conclusions**

The DMFs of $H_2O$ and HDO were retrieved from the spectra observed by the ground-
based high resolution FTIR at Hefei site. Time series of $X_{H2O}$ were compared with
GOSAT data. The mean relative bias was 2.85% and the correlation coefficient is 0.98
between FTIR and satellite date, showing a good agreement. $X_{HDO}/X_{H2O}$ ratio expressed
as $\delta D$ are calculated. $\delta D$ from nearby Tsukuba station with similar latitude are used to
verify the accuracy of our data. It is found that the $\delta D$ in Hefei showed a same trend as
that in Tsukuba, with the maximum value in summer and minimum in winter. Variation
of $\delta D$ ranges from -36.46‰ to -282.3‰, while $\delta D$ in Tsukuba is from -35.74‰ to -
198.37‰.
The relationship of meteorological parameters with stable isotopes of water vapor were
analyzed. The $\delta D$ values and temperature showed an obvious positive correlation, with
the correlation coefficient of 0.83, while $\delta D$ has weak correlation with relative humidity,
with the correlation coefficient of 0.45. $\ln(\delta D*1000+1)$ has obvious correlation with
$\ln(X_{H2O})$, with the correlation coefficient of 0.88.


Further, we used the NOAA HYSPLIT model to calculate the back trajectories of air
parcels in Hefei, and performed the cluster analysis and PSCF analysis. The results of
cluster and PSCF analysis showed the sources of δD and their potential contributions
are mainly from the surrounding area of Hefei site and especially in the east area.
Also, the δD value of evapotranspiration is calculated based on Keeling plot analysis.
δD value of evapotranspiration varied from (15.3 ± 2.9) ‰ to (-114 ± 8.9) ‰, and the
averaged δD value of evapotranspiration is -44.43‰.
The FTIR technique offers a new opportunity to monitor the stable isotopes of water
vapor. The long time series of the stable isotopes of water vapor provide a basis of
revealing the water cycle of the atmosphere. The further research work will focus on
accurate retrieval of $H_2^{18}O$ from solar absorption spectra, and can clearly clarify the
water cycle in combination with HDO.

**Data availability.** The **GFIT** software can be found via https://tccon-wiki.caltech.edu/.
The data used in this paper are available on request.
**Funding sources and acknowledgments.**
We gratefully acknowledge the support of the National Natural Science Foundation
of China (41775025; 41405134; 41575021; 91544212; 41605018), the National Key
Technology R&D Program of China (2017YFC0210002; 2016YFC0200800;
2016YFC0200404; 2016YFC0203302), and Anhui Province Natural Science
Foundation of China (Grant No. 1308085MD79) for conducting this research. We thank
the NIES GOSAT Project Office for the GOSAT TANSO-FTS SWIR $X_{H2O}$ data. The
authors gratefully acknowledge the NOAA Air Resources Laboratory (ARL) for
providing the HYSPLIT transport model (http://ready.arl.noaa.gov/HYSPLIT.php).

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



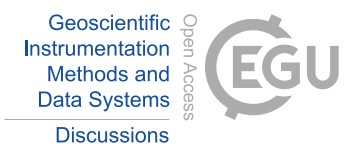

Table 1. The statistics of monthly averaged δD and surface temperature.

| | Sep. | Oct. | Nov. | Dec. | Jan. | Feb. | Mar. | Apr. | May. | Jun. | Jul. | Aug. |
|---|---|---|---|---|---|---|---|---|---|---|---|---|
| δD (‰) | -126.89 | -131.94 | -209.71 | -221.13 | -257.86 | -180.4 | -107.65 | -111.92 | -113.66 | -95.94 | -69.52 | -79.54 |
| Variation amplitude of δD (‰) | 117.5 | 172.46 | 168.64 | 186.38 | 392.17 | 213.66 | 182.29 | 118.7 | 155.85 | 87.76 | 67.9 | 93.78 |
| Temperature(°C) | 30.18 | 24.01 | 14.55 | 8.94 | 4.74 | 11.65 | 16.07 | 24.01 | 26.49 | 31.12 | 37.09 | 34.63 |
| Variation amplitude of temperature (°C) | 10.9 | 15 | 13.9 | 14.1 | 19.5 | 19.2 | 14.4 | 11.4 | 14.4 | 10.5 | 6.3 | 8 |







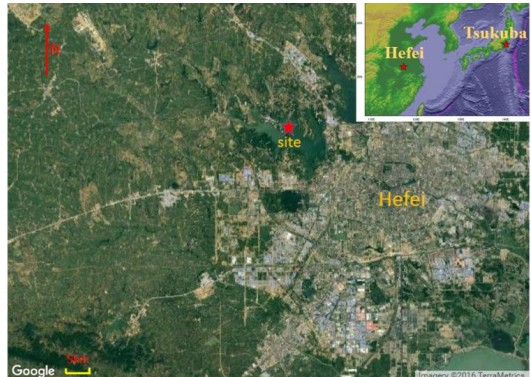


Figure1: Positions of Hefei and Tsukuba sites

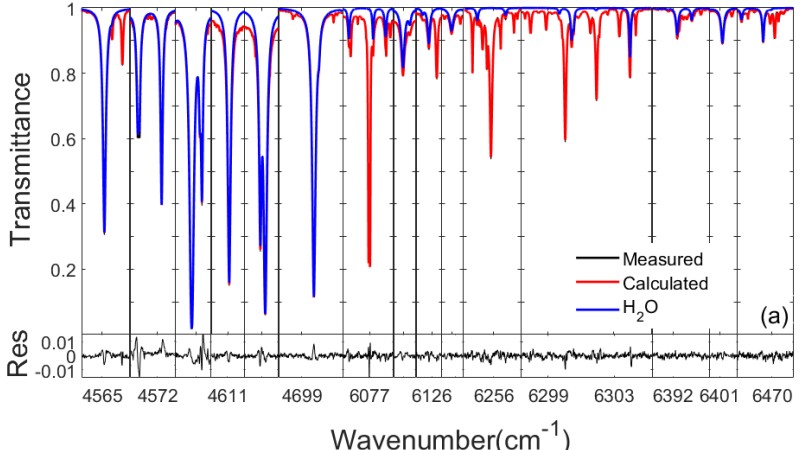


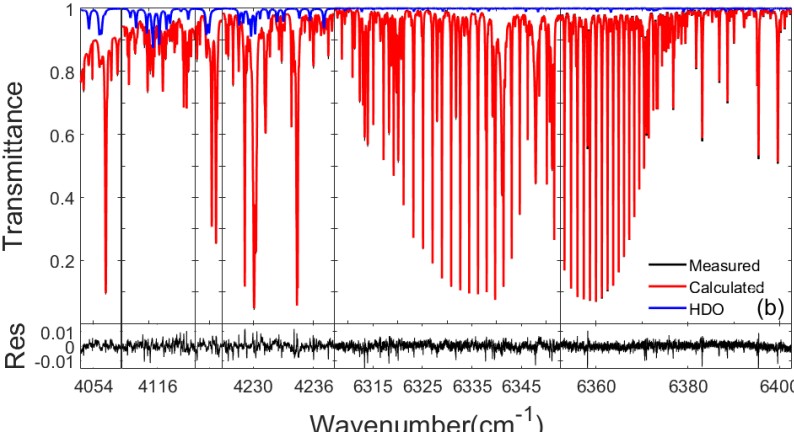




Figure 2: The spectral fitting of $H_2O$ (a) and HDO (b). The black lines represent the measured

spectra,the red lines represent the calculated spectra, the blue lines respesent the absorption signals for

$H_2O$ and HDO. The bottom panels are the spectra fitting residuals.

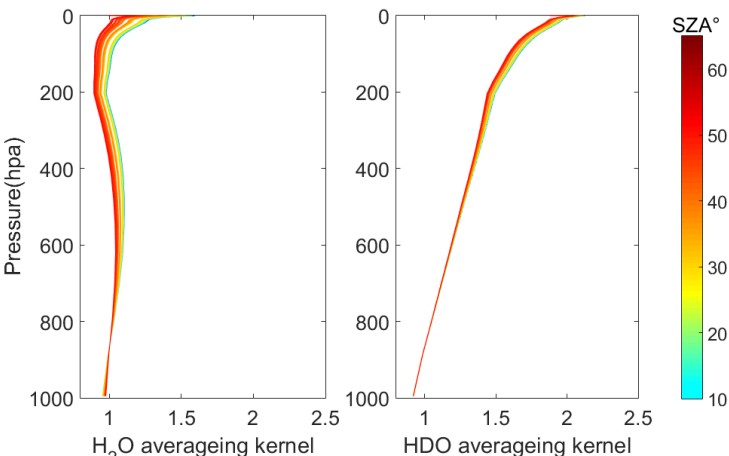


Figure 3: Column averaging kernels of $H_2O$ and HDO

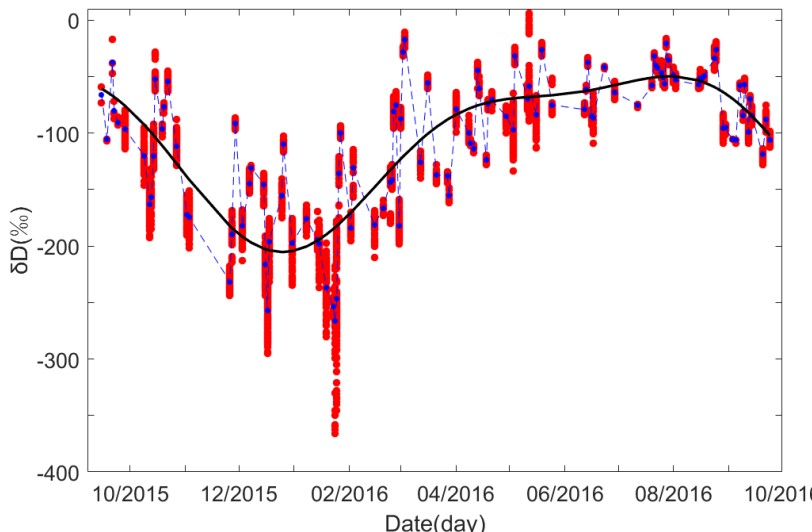


Figure 4: Time series of δD from September 2015 to September 2016 at Hefei site. The red points are
the individual measurements, the blue points represent the daily averaged data, and the black line is the

Fourier fitting line of time series.


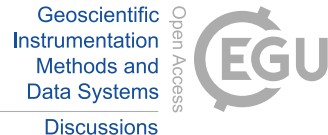



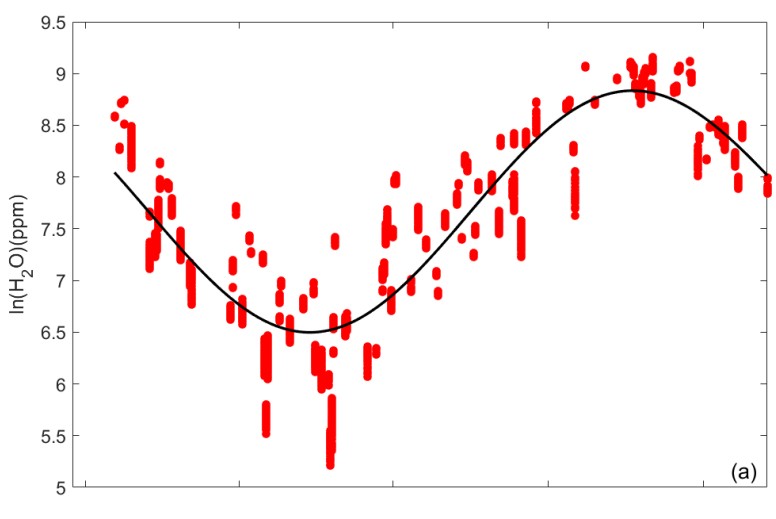


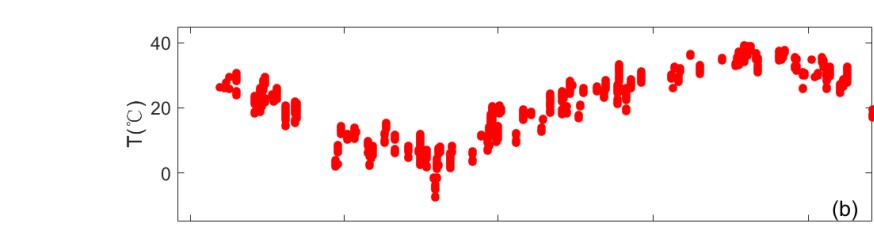

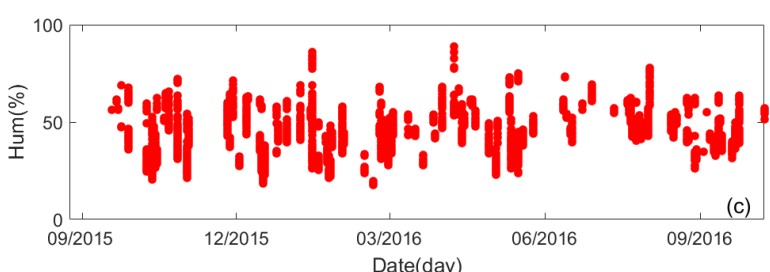

Figure 5: Time series of $X_{H2O}$, surface temperature and surface relative humidity from September 2015
to September 2016 at Hefei site. (a) Time series of $X_{H2O}$ with the ln($X_{H2O}$) of Y axis, and the black line
was fitted line; (b) Time series of surface temperature; and (c) Time series of surface relative humidity.



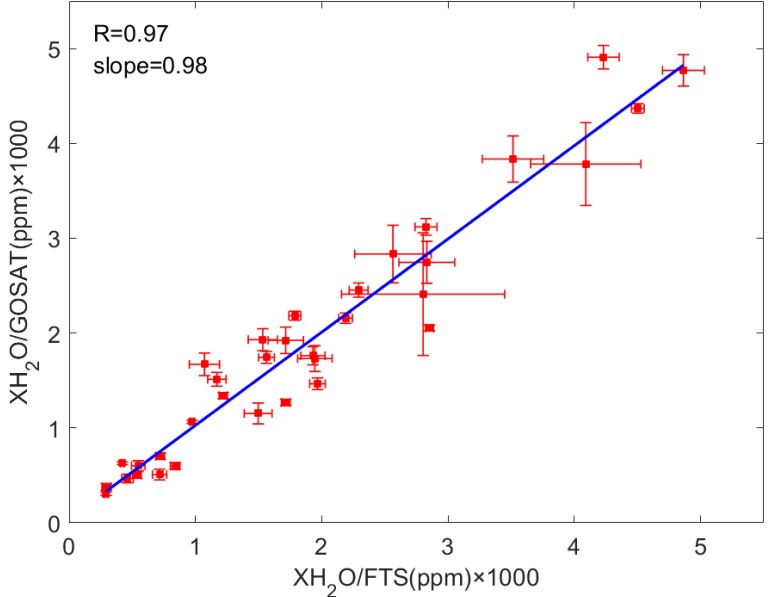


Figure 6: The scatter plot of $X_{H2O}$ at Hefei site and the coincident GOSAT data


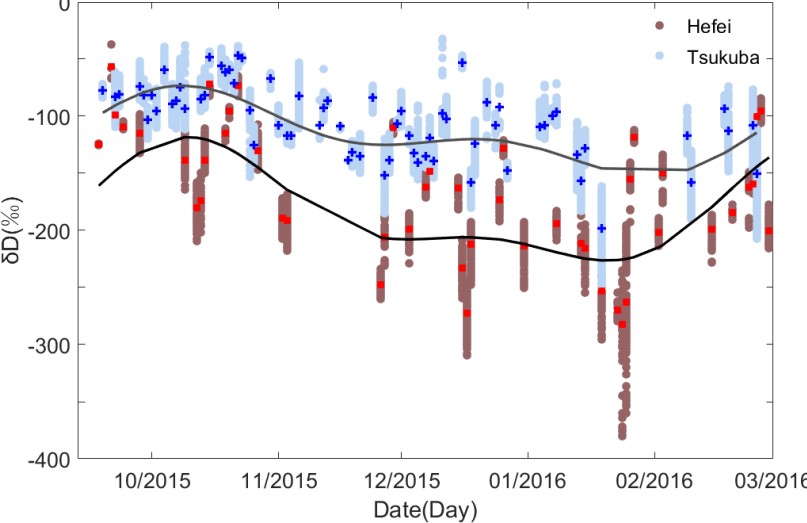


Figure 7: Time series of δD in Hefei and Tsukuba stations, respectively. The red and blue dots are daily

averaged δD at Hefei and Tsukuba, the black lines are the Fourier fitting lines of time series for each

site.


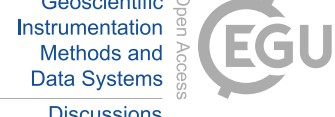


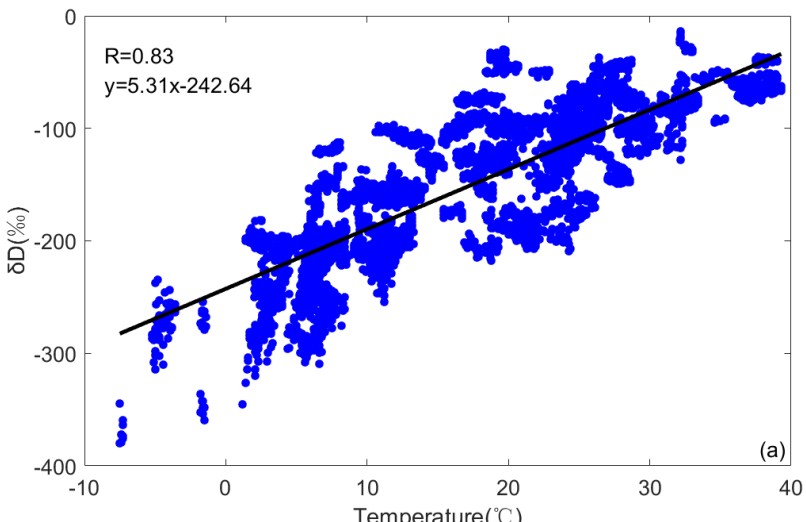


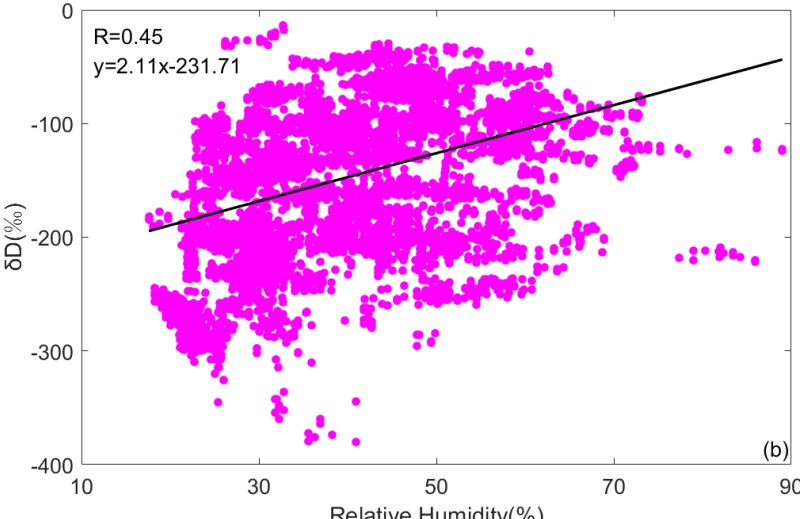






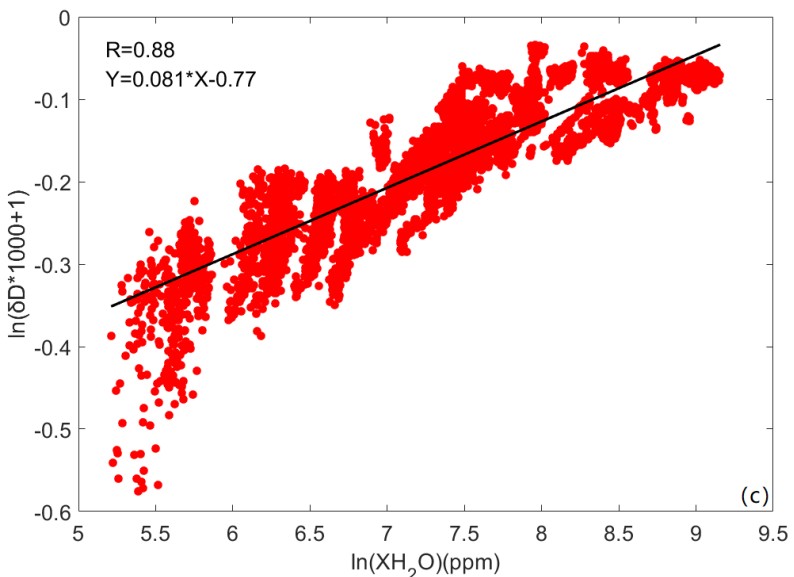


Figure 8: Relationship of the stable isotopes of water vapor with the meteorological parameters. (a).

The relationship between δD and temperature. (b).The relationship between δD and relative humidity.

(c). Scatter plots of ln(δD/1000+1) and ln(X$_{H2O}$)


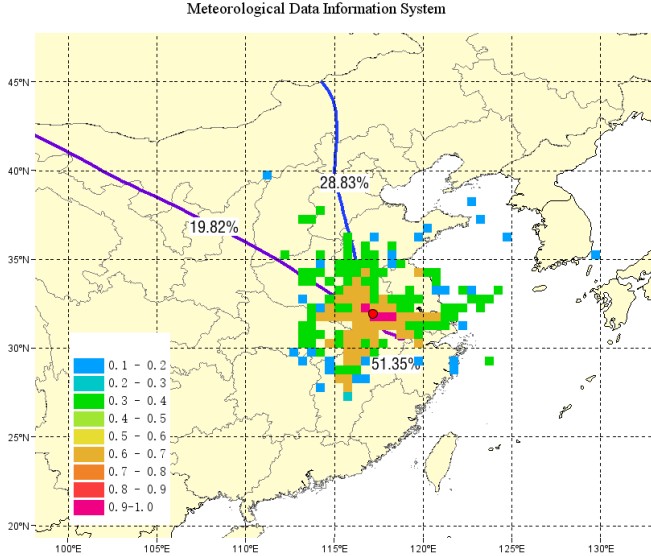


Figure 9: Cluster calculated of backward trajectories and the WPSCF of δD analysis at Hefei. The

colourful area in the map denotes the potential sources regions calculated from the trajectory statistics.





And the colourful line represent the cluster analysis result.


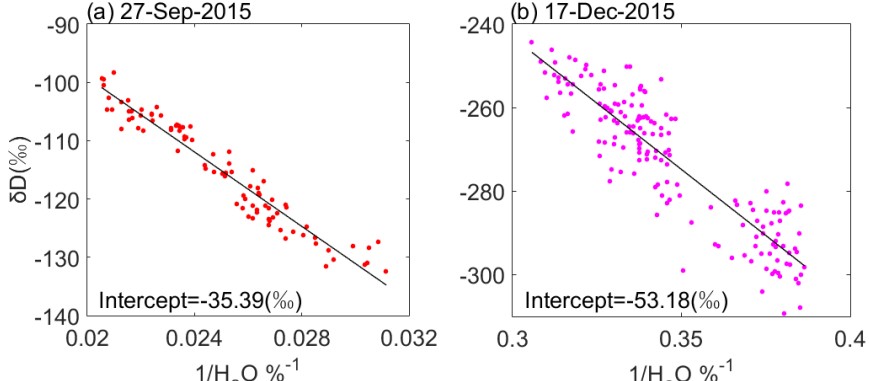


Figure 10: Keeling plots of measurements on October 27, 2015 and December 17, 2015.



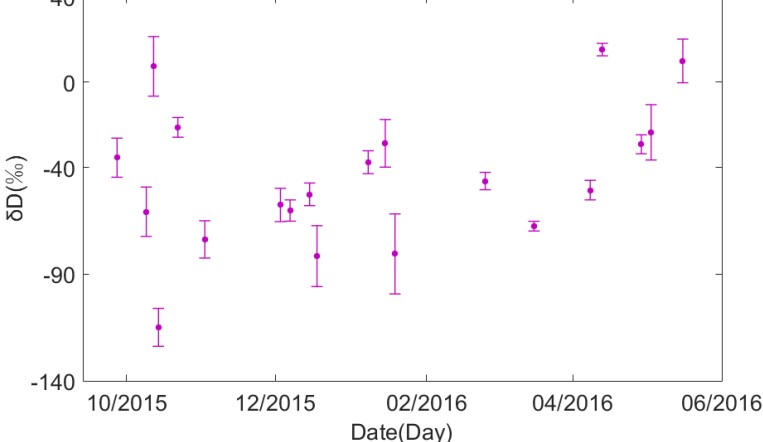


Figure 11: δD values of evapotranspiration during the measurement period. The error bars are standard

deviations of value