# Peer review of "Continuous observation of Stable Isotopes of Water"

_Geoscientific Instrumentation, Methods and Data Systems, 2018_

## Referee Comment (RC1) · Anonymous Referee #1 · 30 Jan 2019

This work made a valuable analysis from the data of the ratio of HDO to H2O ($\delta$D) measured by FTIR in Hefei, China. The dataset was well applied to validate the GOSAT satellite data product of $\delta$D. This work enriches the knowledge of water cycle to the public.

Two specific comments are as follows: (1) The English needs to be improved, particularly for the text of conclusion section. (2) Listed reference papers are suggested to be cut down to some extent.

Some minor corrections are also needed to be made for misspelling words. (1) L151-152, "satellite date" should be "satellite data"; (2) L118,L119: "kernals" should be "ker-

nels"; (3) In Eq. (2), the same size should be kept for the same type of signs.
* * *

---

## Author Comment (AC1) · 21 Feb 2019

Response to comments #1 We appreciate your constructive and positive comments. The comments and proposed corrections have been taken into account and helped to improve the paper. Each comment has been addressed as follows. There is an extensive discussion among the authors regarding how to revise the content.

-Reviewer 1 This work made a valuable analysis from the data of the ratio of HDO to $H_2O$ ($\delta D$) measured by FTIR in Hefei, China. The dataset was well applied to validate the GOSAT satellite data product of $\delta D$. This work enriches the knowledge of water cycle to the public. Two specific comments are as follows:

[Figure]

(1) The English needs to be improved, particularly for the text of conclusion section. Response: We improved the English of the manuscript, particularly in the text of conclusion. Please see the supplement written in red.

(2) Listed reference papers are suggested to be cut down to some extent. Response: We cut down some unimportant reference paper.

Some minor corrections are also needed to be made for misspelling words. (1) L151-152, "satellite date" should be "satellite data"; Response: We changed the "satellite date" to "satellite data". Please see the Lines 149-152, written in red.

(2) L118,L119: "kernals" should be "kernels"; Response: We changed the "kernals" to "kernels". Please see the Lines 117-119, written in red.

(3) In Eq. (2), the same size should be kept for the same type of signs. Response: We kept the same size for the same signs in Eq. (2). Please see the Lines 110-111, written in red.

Please also note the supplement to this comment:
https://www.geosci-instrum-method-data-syst-discuss.net/gi-2018-43/gi-2018-43-AC1-supplement.pdf

---

## Referee Comment (RC2) · Anonymous Referee #2 · 19 Jul 2019

The authors describe measurements of stable isotopes of water at a local site (Hefei) in China and observe seasonal variations in the isotopic ratios. Comparisons are made with data from previously published data from Tsukuba (Japan). The paper is well organised and is written in good English. However, it is not clear what the signifcnce of the papers is. The instrument used is a commercially available instrument and the method of measuring is not new. Apart from giving the wavelegnth range, the instrument is not described at all. Comparisons with satellite measurements are made and a good correlation is found, but as no further data on either the ground based instrument nor the satellite instrument is given it is unclear if these findings are significant, or if the ground based data can be considered as calibration of the space instrument. As

neither the instrument design nor the measurement method seem to be innovative, or are at least not described, the manuscript is outside the scope of the GI journal.

If the results are scientifically valuable, this should be made clear and some statements on the implications should be made. If this can be done the paper can be re-submitted to a journal addressing atmospheric sciences, for example Atmospheric Chamistry and Physics (ACP). The manuscript should be rejected for publication in GI as it is out of scope for this journal.

---

## Editor Comment (EC1) · Maria Genzer (Editor) · 26 Aug 2019

It is not clear what the signifcance of the papers is. The instrument used is a commercially available instrument and the method of measuring is not new. As neither the instrument design nor the measurement method seem to be innovative, or are at least not described, the manuscript is outside the scope of the GI journal.

I do not recommend the authors to re-submit a revised manuscript.

If the results are scientifically valuable, this should be made clear and some statements on the implications should be made, and the paper could be then re-submitted

to a journal addressing atmospheric sciences, for example Atmospheric Chemistry and Physics (ACP).

Interactive comment